# Walking Down the Memory Maze: Beyond Context Limit through Interactive Reading

## Abstract

Large language models (LLMs) have advanced in large strides due to the effectiveness of the self-attention mechanism that processes and compares all tokens at once. However, this mechanism comes with a fundamental issue — the predetermined context window is bound to be limited. Despite attempts to extend the context window through methods like extrapolating the positional embedding, using recurrence, or selectively retrieving essential parts of the long sequence, long-text understanding continues to be a challenge. We propose an alternative approach which instead treats the LLM as an interactive agent, allowing it to decide *how* to read the text via iterative prompting. We introduce MemWalker, a method that first processes the long context into a tree of summary nodes. Upon receiving a query, the model navigates this tree in search of relevant information, and responds once it gathers sufficient information. On long-text question answering tasks our method outperforms baseline approaches that use long context windows, recurrence, and retrieval. We show that, beyond effective reading, MemWalker enhances explainability by highlighting the reasoning steps as it interactively reads the text; pinpointing the relevant text segments related to the query.

## 1 Introduction

Large language models (LLMs) have witnessed significant advancements due to the increased model size, expanded pretraining data, and the adoption of the Transformer architecture with self-attention (Vaswani et al., 2017). As LLMs evolve in capability, users increasingly seek to use longer input sequences during inference. This results in a growing demand in querying for information in long documents, analyzing legal or scientific papers, and managing extended conversational dialogues. These tasks involve consuming a large amount of information, highlighting the importance of longer context processing.

Despite the rapid development, the limitation of the self-attention mechanism becomes apparent as its memory usage increases with longer sequences, consequently limiting the size of the context window. To address this, different approaches have been employed, such as designing lighter and more efficient attention schemes (Zaheer et al., 2020), finetuning with extrapolated or interpolated positional embeddings (Press et al., 2022; Chen et al., 2023), incorporating recurrence to bring forward information from preceding text segments into the next (Rae et al., 2019; Fan et al., 2020; Xu et al., 2022), or retrieving relevant parts of the text (Lewis et al., 2020; Izacard & Grave, 2020). However, these approaches are still limited by design. The context window, no matter how long it is extended, assumes a fixed size, and not all positions within it hold equivalent significance (Liu et al., 2023). While recurrence can manage infinite-length sequences, it often misses out on retaining information from earlier segments. Additionally, retrieving segments from the coherent long-text might be ineffective, given that many retrieval systems are tailored to distinguish similar but distinct documents (Chen et al., 2017).

To address these issues, we develop a fundamentally different approach which treats the model with a finite context window as an interactive agent, rather than simply processing the entire sequence in one go. To this end, we introduce MemWalker, a method that enables the model to read the long-text interactively via iterative LLM prompting. MemWalker operates through a two-stage approach: 1) *memory tree construction* and 2) *navigation*. During the first stage, the long-text is segmented into small chunks that fit within the LLM's context window. The LLM then subsequently

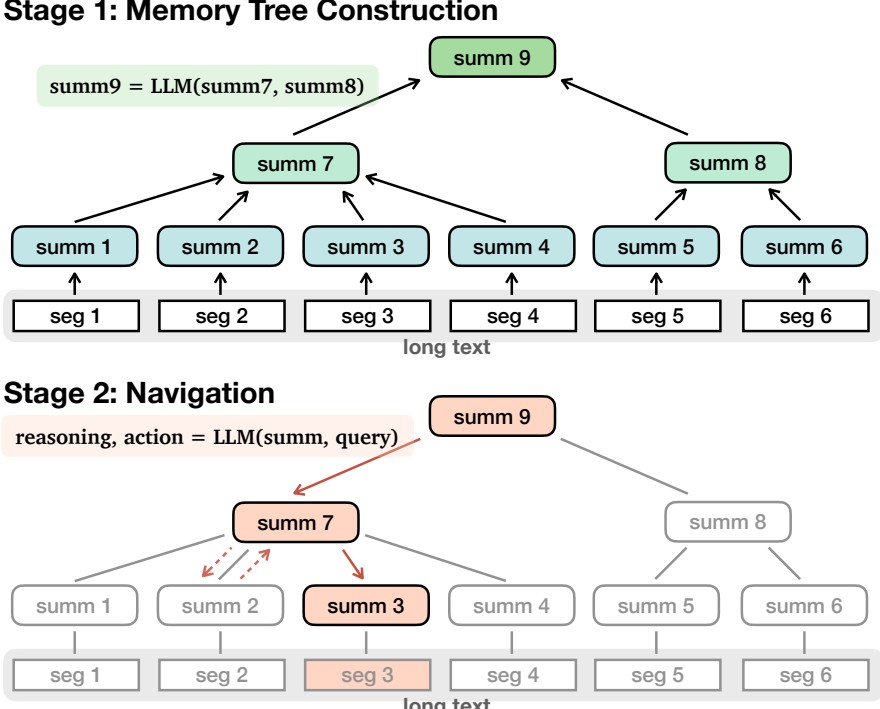

Figure 1: The two-stage procedure of MEMWALKER. Top (stage 1): the memory tree is constructed. The long text is split into segments of a predetermined size and each segment is first summarized into a summary node. The summary nodes are recursively summarized into higher level nodes until it reaches the root. Bottom (stage 2): Given a query, the LLM navigates the tree structure via iterative prompting and finds the node that contains relevant segment to form the answer. At each node, the LLM decides the action by first reasoning about the child summary nodes by sampling from the distribution $\text{LLM}(reasoning, action \mid summ, query)$. The LLM can choose the *revert* action to return to the parent node if it chose the wrong path or the segment at hand is irrelevant (dashed red arrow). See Table 1 for a detailed example showing the LLM prompts that enable navigation.

summarizes each segment into a textual summary node. These summary nodes are progressively further summarized into higher-level summary nodes, thus building a tree structure (Figure 1). To answer a user query, the LLM begins navigation from the tree's root node. It traverses the tree, inspecting various parts of the text to identify the path and segment relevant to answer the query. As a result, MEMWALKER can go beyond the context limit, efficiently processing texts and localizing the important segments of the long-text, without additional finetuning.

We evaluate MEMWALKER on three long context question answering tasks and show superior performance against recurrence, retrieval, and vanilla LLM baselines. MEMWALKER also outperforms other open long context systems that can take $8,000$ to $16,000$ tokens. We provide an analysis of the effectiveness of MEMWALKER, and show it can reason about navigation decisions, incorporate working memory during traversal, and recover from errors made in early navigational steps.

## 2 RELATED WORK

**Context window scaling.** A straightforward approach to enable a longer context sequence is to tune the pre-trained language models and extrapolate their positional embeddings on longer text sequences (Press et al., 2022; Chen et al., 2023). Another direction is modified self-attention (Beltagy et al., 2020; Zaheer et al., 2020; Guo et al., 2022; Ainslie et al., 2023). This approach has advanced in large strides thanks to training techniques such as Flash Attention (Dao et al., 2022) that greatly reduce the memory footprint. Despite the recent advances, this approach comes with two natural limitations: 1) to enable models to take in longer sequences, the model needs to be fine-tuned,

incurring a non-negligible cost and 2) the attention mechanism may become less effective due to positional biases as the sequence length becomes very long (Liu et al., 2023).

**Recurrence.** Recurrent architectures have been extensively studied to tackle long sequence problems, from recurrent neural network based models Hochreiter & Schmidhuber (1997); Miller et al. (2016) to the modern Transformer based models (Dai et al., 2019; Rae et al., 2019; Fan et al., 2020; Xu et al., 2022; Bulatov et al., 2023; Chevalier et al., 2023). However, each recurrence step incurs information loss and the training objective does not guide "how to compress" with regard to downstream tasks. Typically this compression means that recall of older sequence information is weaker compared to recent information.

**Retrieval.** Retrieval systems are commonly used to select relevant documents from a large pool of documents, and have been incorporated into neural models in various ways (Chen et al., 2017; Dinan et al., 2018; Lewis et al., 2020). For long sequence reading, retrieval based methods typically first embed the text segments into vector representations and retrieve them based on the query instead of feeding the entire sequence into the model such as in Fusion-in-Decoder Izacard & Grave (2020) or kNN variants that attend to external memory such as Memorizing Transformers (Wu et al., 2022).

**Reasoning agents.** Instead of taking the long text as a single monolithic input, a model can act as an agent that reads part of the text and takes flexible actions. Work such as WebGPT (Nakano et al., 2021) and WebShop (Yao et al., 2022) allow the model to scroll through the internet and search for the requested answer or item. While their atomic actions allow for interactive search for relevant content, the models were not designed for understanding long and coherent texts. On the other hand, PEARL (Sun et al., 2023) prompts the model to generate pseudo APIs for the model to call in order to focus on the right parts of the long text. However, the method operates within the LLM's context window, rather than being a memory-access approach that goes beyond the context limit. Other works leveraged iterative prompting to reason and plan for long text generation tasks such as Re3 (Yang et al., 2022) and RecurrentGPT (Zhou et al., 2023). Self-Notes (Lanchantin et al., 2023) interleaved self-generating notes and the input data to perform better reasoning. Prior to current LLMs, LSTMs were also applied to searching through document structures (titles, subsections) Geva & Berant (2018). Recursive tree structure has also been explored in the context of summarization of long text such as books in (Wu et al., 2021), but was not used for memory navigation in that work. Llamaindex (Liu) provides practitioners who aim to use LLMs as building blocks to quickly build applications. In particular, the Tree Index shares similarity to MemWalker in terms of using trees to process documents. We operatoinalize and extend this idea by analyzing the the reasoning capability threshold and the necessity of working memory go beyond simple tree building and querying and focuses on how this leads to better models to understand long context.

## 3 MEMWALKER: AN INTERACTIVE READER

We study tasks related to long-context question answering – given a long-text $x$ and a query $q$, the model aims to generate the response $r$.

MEMWALKER follows two steps: 1) *memory tree construction*, where the long-context is broken down into a tree data structure. This construction does not depend on the query, and can hence be computed in advance if the sequence data is available beforehand. 2) *navigation*, in which the model navigates this structure upon receiving a query, gathering information to craft a suitable response. MEMWALKER assumes access to an underlying LLM, and both construction and navigation are achieved through iterative LLM prompting.

**Memory tree construction.** MEMWALKER first creates a tree data structure, $\mathcal{T}(x)$, from the long-text $x$. Each node is represented by text that encapsulates the summaries of all its child nodes below it. Specifically, the long-text $x$ is divided into segments $(c_1, \ldots, c_n)$. The LLM then summarizes each segment into a summary at the first level, represented as $s_i^{l=1} = \texttt{LLM}(c_{\leq i})$, $i = 1..n$. The initial summary nodes are subsequently summarized further into higher level nodes, $s_j^{l+1} = \texttt{LLM}(s_i^l, \ldots, s_{i+M_t}^l)$ where $M_t$ denotes the number of nodes in the $t$-th grouping at level $l$. This process continues until the topmost root node, $s^L$ is generated. The complete tree generation

Table 1: Example trajectory from the QuALITY dataset. The LLM first sees the content of the children nodes at the root node (`summ 9` in Figure 1) and generates the response (takes action 0 to enter `summ 7`). When arriving at the leaf node (`summ 2`), the LLM determines that there is not enough information, therefore takes the action to revert (action -1) to the parent node. After hopping back-and-forth between nodes, the LLM commits to a Leaf node (`summ 3`) and answers the question. Yellow indicates triage prompt and purple indicates leaf prompt described in §3. Text after // denotes comments that are not processed by the LLM.

| Navigation Trajectory |
| --- |

**`summ 9`** — The following passage(s) are the summaries of the different parts of a story.
To answer the question: *Why did Ro change his mind about the people on Mars being backwards?*
Which of the following summary is MOST LIKELY to contain information about the answer?
First provide reasoning to compare the summaries before you make the decision.

Summary 0: The story is set on Mars and follows the adventures of Ro, [...] // `summ 7`
Summary 1: Ro, a young Martian, is climbing down a cliff to rescue [...] // `summ 8`

Relpy with the passage number as your action.
You MUST choose one summary number and you should reply with the following format:
################################
Reasoning: ...
Action: 0 / 1 / 2, ...
################################

**Response** — Reasoning: Summary 0 is most likely to contain information about why Ro changed his mind about the people on Mars being backwards, as it mentions Ro's interactions with the people from Earth and their advanced method of communication.
Action: 0

LLM enters `summ 7` then enters `summ 2`

**`summ 2`** — Read the text in triple quotes and answer a question:
Story background information: Ro, a young Martian who has returned from his travels to find his home overrun by the Oan, a tribe of rat-like creatures. [...] // [WORKING_MEMORY] carried from `summ 7`
Main text: In his left hand and under his armpit Ro carried stones. They were of a good weight and would make short work of any Oan who was foolish enough to cross his path. [...] // `seg 2`

Why did Ro change his mind about the people on Mars being backwards?
(A) He realized that despite human's technological advancements, they have over-complicated marriage.
(B) He realized that while the humans are physically vulnerable without their weapons, the red people have formidable strength in their arms.
(C) He realized that human males suppress public affection when they are intimidated by other males, whereas male Martians don't hide their affection.
(D) He realized that male humans were petty and even brute when it came to rivalry over women, whereas male Martians were much more civilized.
If the answer CANNOT be inferred from the text above, reply with action -1.
If the answer CAN be inferred from the text above, reply with action -2, and also provide your reasoning, and the final answer.
You are ONLY allowed to reply with action -2 or -1.
Your should reply with the following format:
################################
Reasoning: ...
Action: -2 or -1
Answer: (A) ...
################################

**Response** — Reasoning: The text does not explicitly mention Ro changing his mind about the people on Mars being backwards. Therefore, the answer cannot be inferred from the text.
Action: -1

LLM reverts to `summ 7` then enters `summ 3`

**`summ 3`** — [...] Main text: "Last night I thought that we on Mars are backward. Now I'm not so sure." [...] // `seg 3`
**Response** — Reasoning: Ro initially sees Earth's customs as backward compared to Mars, However, after discussing [...]
Action: -2
Answer: (A)

process is illustrated in Figure 1. Summarization is performed using LLM prompting. We include the prompts for memory tree construction in Appendix A.1.

**Navigation.** The language model, upon receiving a query $q$, starts from the root node $s^L$ to navigate the tree to generate the response $r$. At node $s^l$ that the LLM traverses, it observes the summaries of the nodes one level below $\{s_i^{l-1}, \ldots, s_{i+M_t}^{l-1}\}$. The LLM decides among $|M_t| + 1$ actions — choosing one of the child nodes to further inspect, or to revert to the parent node. At leaf node $s_i^{l=1}$, the LLM can decide one of two actions: *commit* to the leaf node and respond to the query or *revert* to the parent node ($s_j^{l+1}$) if the information in the leaf node (i.e., $c_i$) is insufficient. To make a navigation decision, we can also ask the LLM (via prompting) to first generate a reason in natural language to justify the action, followed by the action choice itself. Specifically, at each node, the model generates a response $r \sim \texttt{LLM}(r \mid s, q)$ where the response is either of the two tuples: 1) $r = (reasoning, action, answer)$ when the LLM is at a leaf node or 2) $r = (reasoning, action)$ when the LLM is at non-leaf nodes.

**Navigational prompt design.** We enable LLM navigation through zero-shot prompting. Our method requires two types of prompt: 1) *triage prompt* and 2) *leaf prompt* (highlighted in Table 1). Triage prompt contains the the query, the summaries of the children nodes, and instructions for the LLM to follow. Triage prompt is used at non-leaf nodes. Leaf prompt contains the content of the segment, the query (and options), and instructions that ask the LLM to either generate the answer or revert to the parent node. Both the triage prompt and leaf prompt specify an output format that the LLM needs to follow. Failure to conform to the format results in invalid actions and the LLM is required to regenerate. If the LLM fails to generate parsable output three consecutive times, the navigation terminates and returns "no answer".

**Working memory.** As the LLM traverses the tree, it can keep information throughout the navigation trajectory and add it to the context. Formally, the LLM generates the response $r \sim \texttt{LLM}(r \mid s, q, m)$ where the extra working memory $m \in \{\emptyset\} \cup \{(s_i, s_{i+1}, \ldots)\}$ is either empty or consists of contents from previously visited nodes. We truncate the working memory such that they can fit in the LLM's context window.[*] Table 1 illustrates the way working memory is added via [WORKING_MEMORY] in the prompt.

## 4 EXPERIMENTAL SETUP

### 4.1 DATASETS & EVALUATION

We use three datasets: QuALITY, SummScreenFD, and GovReport from the SCROLLS benchmark (Shaham et al., 2022). We report accuracy for all datasets.

**QuALITY.** QuALITY is a multiple choice question answering dataset collected by Pang et al. (2022). The dataset contains long-form stories sourced from Project Gutenberg and questions annotated by human annotators. We use a subset of 187 examples for our experiments.

**SummScreenFD.** SummScreenFD (Chen et al., 2022) is a dataset of TV and movie scripts in the form of dialogues among actors originally designed for summarization. We repurpose the dataset into a question answering task where the original provided ground truth summary text is used to generate a "who" question using Stable Beluga 2, with answers then checked by a human expert. The question paired with the original long text becomes the repurposed QA task of 306 examples.

**GovReport.** The GovReport dataset aggregates documents from Congressional Research Service and the U.S. Government Accountability Office together with summaries provided by experts (Huang et al., 2021). We repurpose the dataset into a question answering dataset of 101 examples the same way as for SummScreenFD.

All three datasets feature long contexts per example of varying length – some shorter examples, and some longer sequences. We therefore both report results on the original dataset, and also report on a subset of each task containing only longer sequences, to better evaluate memory access in the harder, longer context case. The thresholds are above 8, 000 tokens for QuALITY, 6, 000 tokens for SummScreenFD, and 12, 000 tokens for GovReport.

---

[*]Further summarizing the working memory as it accumulates would be an alternative approach, which we have not explored in this study.

Table 2: Results on the three question answering tasks, reporting test accuracy. Orig. denotes using the entire dataset and Long denotes the subset of longer sequences. Top: comparison to open long context models. Bottom: baselines and MEMWALKER performance, with all methods using the underlying Stable Beluga 2 LLM with a maximum 4,096-token context length. MEMWALKER outperforms all other systems on longer sequences.

| | QuALITY Orig. / Long | SummScreenFD Orig. / Long | GovReport Orig. / Long |
|---|---|---|---|
| MPT 13B (8k) | 44.4 / 47.3 | 65.0 / 63.5 | 44.6 / 43.8 |
| LongChat 13B (16k) | 43.3 / 48.4 | 62.4 / 61.1 | 54.5 / 52.1 |
| Recurrence | 51.3 / 56.0 | 47.7 / 45.4 | 35.6 / 33.8 |
| Retrieval | 63.1 / 64.8 | 63.7 / 62.2 | 54.0 / 52.1 |
| Full Context (keep left) | 56.7 / 64.8 | 62.7 / 62.7 | **59.4** / 56.3 |
| Full Context (keep right) | **70.1** / 72.5 | 64.7 / 63.1 | 50.5 / 50.0 |
| MEMWALKER | 67.4 / **73.6** | **67.3 / 64.5** | **59.4 / 60.4** |

## 4.2 MODEL

We use Stable Beluga 2 (Mahan et al.) as the base LLM for the majority of our experiments, as it provides state-of-the-art performance compared to several other LLM variants, as we will show. Stable Beluga 2 is an instruction-tuned model built on top of 70B LLaMA-2(Touvron et al., 2023), where the finetuning does not overlap with our evaluation tasks. It has a maximum 4,096 token context length. We use the model in a zero-shot prompting fashion without further fine-tuning or in-context few shot examples for our tasks. We use top-$p$ sampling for both memory tree construction as well as generating action and reasoning for navigation. We set the maximum number of nodes $\max_t M_t = 8, 5, 8$ and segment size $|c| = 1000, 1000, 1200$ for QuALITY, SummScreenFD, and GovReport respectively.

## 4.3 BASELINES

We compare with three baselines memory techniques all based on the same underlying LLM, Stable Beluga 2: 1) full context window, 2) recurrence, and 3) retrieval. The full context window baselines utilize the full 4,096 tokens to process both the long input text and generation. Since the instances in the dataset often exceed the context limit, we perform truncation of the length to the right (most recent) or left (least recent) of the text as the input, as evaluate both approaches. For retrieval, we use Contriever (Izacard et al., 2022) to select segments from the long context based on the query. The highest scored segments are concatenated as the input context to the LLM until they fill the context. Finally, we implement a baseline that recurrently carries information from previous segment tokens to the current one through summarization (Xu et al., 2022), where each segment is 2,500 tokens and the maximum summary size is 500 tokens.

## 5 RESULTS & ANALYSIS

**Main results.** Table 2 shows comparisons between MEMWALKER and other baselines. MEMWALKER outperforms both the recurrence baseline across all tasks by a large margin. This shows the limitation of recurrence, where relevant information to the query is lost after several steps. MEMWALKER also outperforms retrieval where the segments are from a coherent long story instead of separate documents. On these tasks, the full context baselines can perform well in the "Original" task setting, which can contain relatively shorter sequences, although choosing either left or right truncate for best performance seems to be dataset dependent. Still, MEMWALKER achieves higher performance in the Original setting against the Full Context baselines except for the keep right variant on QuALITY and the keep left variant on GovReport, likely due to the positional bias in the dataset where relevant segment often appears at the beginning or the end of the text. However, on the Long version of all three tasks MEMWALKER outperforms all baselines, that is it shows strong performance when memory access becomes more critical. MEMWALKER also outperforms other publicly available models, including LongChat (Li et al., 2023) and MPT (MosaicML, 2023).

Table 3: MEMWALKER performance using different underlying LLMs with different reasoning capabilities, and an ablation on their reason justification component when making a navigation decision ("w/o reasoning" simply predicts the action, with no reason generated, see e.g. Table 1). Valid Action shows the percent of generated actions that are a valid navigation action. We find that the strongest performing LLM (Stable Beluga 2) benefits from reasoning with improved accuracy, while weaker performing LLMs do not (get worse in terms of accuracy and valid actions).

|  | QuALITY Acc. / Valid Action (%) | SummScreenFD Acc. / Valid Action (%) | GovReport Acc. / Valid Action (%) |
|---|---|---|---|
| LLaMA 2 Chat (13B) | 39.6 / 73.2 | 20.9 / 75.5 | 15.8 / 69.0 |
| w/o reasoning | 48.1 / 97.4 | 25.8 / 95.8 | 21.8 / 93.1 |
| LLaMA 2 Chat (70B) | 52.0 / 86.1 | 55.6 / 99.5 | 41.6 / 97.8 |
| w/o reasoning | 59.9 / 100.0 | 58.5 / 100.0 | 42.6 / 100.0 |
| Stable Beluga 2 (70B) | 67.4 / 92.5 | 67.3 / 95.1 | 59.4 / 97.0 |
| w/o reasoning | 66.8 / 100.0 | 64.1 / 90.5 | 52.5 / 98.2 |

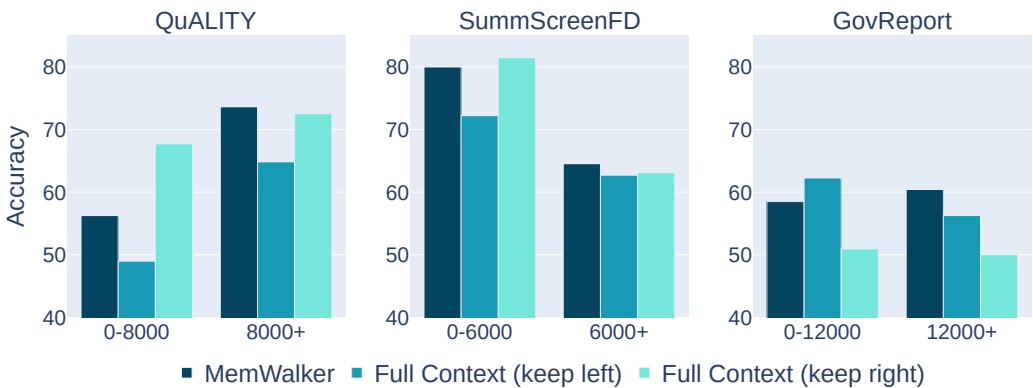

Figure 2: Performance breakdown by context length (in tokens). Each dataset is thresholded into two bucket of equal sizes. MEMWALKER outperforms full context baselines (truncated either left or right, when the sequence does not fit) on longer context sequences, for all three tasks.

**MEMWALKER improves performance on long sequences.** We provide a breakdown of performance by input sequence length for each task in Figure 2. MEMWALKER is not advantageous over Full Context (with truncation left or right) baselines when the text length is short, but outperforms both types of truncation for all tasks for longer sequences. The benefit of interactive reading emerges after the text length is suitably large, i.e. showing better performance once the sequence length is sufficiently larger than the LLM context length of 4, 096.

**Reasoning capability is essential for memory tree navigation.** The effectiveness of MEMWALKER is highly dependent on the underlying LLM's reasoning capability. For each navigation decision, we employ an LLM prompt that requires the LLM to first generate a reason in natural language that justifies the following predicted action, see Table 1. We show in Table 3 how reasoning impacts performance by comparing Llama 2 Chat (13B and 70B parameter variants) and Stable Beluga 2 (70B) with and without the reasoning justification by removing the line "First provide reasoning ... before you make your decision" from the prompt. With the smaller, less capable models (13B), the performance lags behind 70B models by a large margin due to its inability to follow instructions. In fact, asking for reasoning justifications for weaker models *decreases performance*, presumably due to their inability to generate and make use of such reasons. Stable Beluga 2 outperforms Llama 2 Chat for the same LLM size, and also displays heightened reasoning ability. For Stable Beluga 2, asking for reasoning justification *improves performance* across all tasks. This highlights the main characteristic of MEMWALKER: if an LLM passes a critical reasoning ability threshold, it can reason about a long input in multiple rounds without errors cascading quickly across rounds. For weaker LLMs that cannot make good navigation decisions, errors could compound and

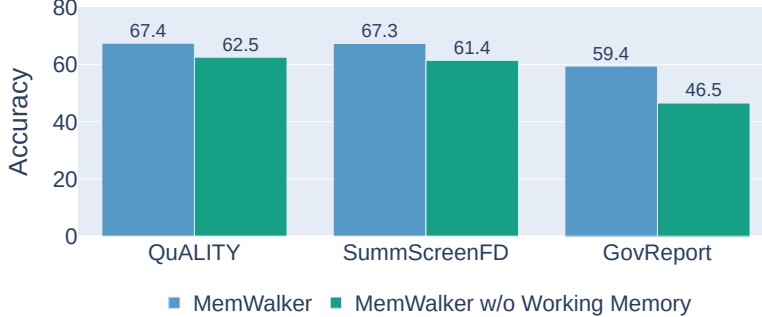

Figure 3: MEMWALKER performance comparisons between using working memory and without (i.e., the LM only looks at the content of the children memory tree nodes, rather than memory from all the nodes it has traversed). Inclusion of working memory yields large gains.

Table 4: MEMWALKER navigation analysis. Stray ratio: percentage of paths that contain the *revert* action. Recovery Rate: percentage of stray paths that recover and answer the query correctly.

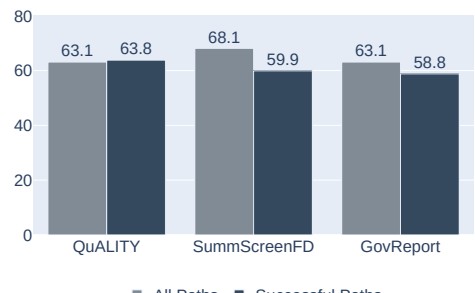

|              | Stray Ratio | Recovery Rate |
|--------------|-------------|---------------|
| QuALITY      | 15.0        | 70.0          |
| SummScreenFD | 18.6        | 59.6          |
| GovReport    | 18.8        | 79.0          |

Figure 4: Percentage comparison of total tokens processed against the tokens of the original example between all paths vs. successful paths.

overall performance suffers. As LLMs will only improve in reasoning ability over the coming years, we expect methods like MEMWALKER will become more and more effective.

**Navigating the memory tree requires working memory.** As MEMWALKER makes decisions to traverse the memory tree and read relevant segments, it might lose sight of the overall context. The model thus carries information from the nodes along the navigation path as working memory, where the content of the working memory updates as the model selects the next path. We evaluate the performance of MEMWALKER with and without working memory, with results given in Figure 3. We find a significant performance degradation without working memory across all tasks, with a 5–13% drop in accuracy, showing the importance of this component.

**MEMWALKER can recover from stray paths.** As MEMWALKER navigates the memory tree, it needs to not only find the path towards the most pertinent segments, but also potentially to recover from traversal errors should they occur. We report recovery statistics in Table 4. MEMWALKER executes a revert navigation action (and hence changes path) for around $15\% - 20\%$ of examples, but of those examples can recover and get those examples correct $70\%$ of the time for QuALITY, $\sim 60\%$ for SummScreenFD, and $\sim 80\%$ for GovReport.

**MEMWALKER enables efficient reading.** Since MEMWALKER determines which parts of the long text it needs to read, the effective content that needs to be read may be smaller than the entire sequence. We report the percentage of the long context read averaged over all examples, for each of the three tasks, in Figure 4. We find that between only 63%-69% of the text on average needs to be read to answer the question including the content of the tree nodes. Among successful paths, the reading required further reduces to 59% - 64%.

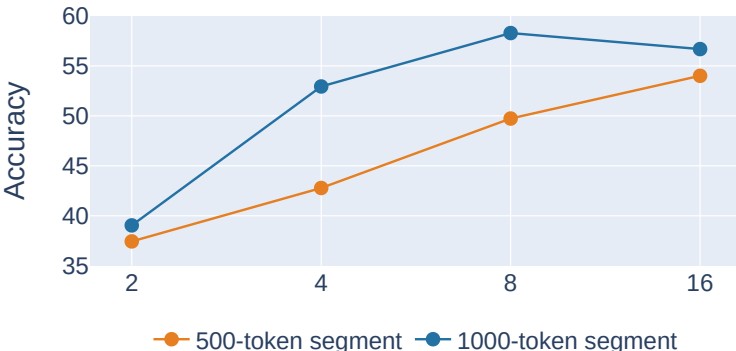

Figure 5: Performance trade-off of different memory construction configurations on QuALITY. x-axis: maximum number of nodes that can be connected to a parent node. Red: summarizing $1,000$-token segments. Blue: summarizing 500-token segments.

**Memory tree construction trade-off.** A fundamental trade-off arises as we construct the memory tree — summarizing larger segments compresses more information into a node to reduce the depth of the tree, but risks losing fidelity of the content. Similarly, connecting many lower level nodes to the upper one can help flatten the tree, yet render the navigation task harder for the LLM at each node. Figure 5 shows the performance of different configurations of the memory tree on QuALITY. Summarizing larger segments is generally more beneficial than smaller segments as well as connecting more children nodes to the parent. However, the performance plateaus as the maximum number of nodes increases, showing the trade-off with respect to how much information can be packed into the nodes during memory tree construction.

## 6 CONCLUSION

We propose MEMWALKER, an interactive reading agent which uses iterative LLM prompting to decide which part of the content should be read closely based on its own reasoning. Our approach first builds a structured memory given long context sequence data, and then makes navigation decisions of the pertinent parts to read given a query. Our method shows superior performance against a number of baselines including various long context length models, retrieval and recurrence baselines, in particular for longer sequence tasks. Detailed analysis highlights a number of important factors, including our method's ability to reason about navigation decisions, ability to revert navigation to a different path when necessary, and incorporation of a working memory. Future work should explore many new directions that MEMWALKER opens up, in particular its application to different data structures other than trees, and finetuning its performance specific to the interactive reading goal.

## 7 LIMITATIONS

MEMWALKER exhibits three major limitations. First, the memory tree generation might not scale too well if the sequence's length becomes extremely long. The increase in sequence length entails more nodes in the tree and hence renders the tree construction process onerous. Workaround such as trading off the granularity of the summary in exchange for speed might be viable. Nonetheless, the issue of scaling remains a limit. In this setting it may make sense to generalize MEMWALKER to a combination of tree and hash Bawa et al. (2005) or other alternative data structure, whilst retaining its traversal ability via LLM prompting. Second, MEMWALKER only works when the LLM exhibits a strong enough reasoning capability, which according to our experiments is required to be large (over 70B) and instruction-tuned. If the reasoning capability falls short, the error compounds and the method would fail. Enabling a smaller model that can perform a similar instruction following procedure could be useful for scaling the method. This could be made possible by removing the following third limitation. Third, MEMWALKER only uses zero-shot prompting and does not leverage fine-tuning to further improve the interactive reading capability. This could be done, for example, by performing interactive reading and collect the successful paths for further fine-tuning.

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
