# A APPENDIX

## A.1 PROMPTS

We provide full prompts for both memory tree construction and navigation described in §3.

### A.1.1 MEMORY TREE CONSTRUCTION PROMPTS

We use two prompts for memory tree construction (the construction component of Table 5). The first (leaf) instructs the LLM to summarize the text segment into a comprehensive summary. After this step, the segments are grouped and summarized into non-leaf node summaries. The summaries ([CHILD_SUMM_NODE_0], [CHILD_SUMM_NODE_1],..., [CHILD_SUMM_NODE_N]) are grouped and concatenated as the summary content of their parent node. During this process, if the concatenated summaries exceed the predetermined length, the second construction prompt is used to further summarize the text (i.e., [SUMMARIES]) for the parent node.

### A.1.2 NAVIGATION PROMPTS

We use two navigation prompts (triage and leaf) as described in §3. We show the general prompt template in the navigation stage of Table 5.

## A.2 EXAMPLES

We provide an extra navigation example in Table 6.

## A.3 FLAT LEAFS BASELINES

We provide baselines which only uses the leaf nodes to make the prediction. We prompt the models to first generate whether the answer can be inferred from the summary leaf node. If the answer is yes, then keep generating the answer. We take the majority vote across all generations to determine the final answer. The results on QuALITY is shown in Table 7. We observe that the flat summary approach gives a pretty unstable result. This is mainly due to ill-calibration, i.e., the model tends to predict overwhelmingly yes or no given the text (Stable-Beluga 70B gives overwhelmingly no as answer). This suggests that using the tree structure is necessary in the zero-shot scenario.

## A.4 RETRIEVAL BASELINES WITH VARYING BLOCK SIZES

We provide retrieval baselines results as suggested on QuALITY with varying retrieved number of chunks $k$ in Table 8. Results show a consistent trend that increasing the number of retrieved chunks improves performance. The issue of distraction by irrelevant content has not been present in our setting.

## A.5 FINISH RATIO

We provide finish ratios on each dataset where the model is unable to parse its response 3 consecutive times. As shown in Table 9, on all datasets, finish ratios are above 90% and are not the major source of error in MemWalker.

Table 5: Prompts used for the memory tree construction stage and the navigation stage. For the memory construction stage, `[TEXT_OF_SEGMENT]` is filled with the segment text at the leaf nodes. `[SUMMARIES]` is the concatenated summaries from the child nodes and will be further summarized if it exceeds the predetermined length. For navigation, `[QUERY]` is the query, `[OPTIONS]` are the multi-choice options (only in QuALITY), `[CHILD_SUMM_NODE_n]` represents the summary text of the $n$-th child node, and `[WORKING_MEMORY]` is the information carried from previous nodes. Yellow indicates triage prompt and purple indicates leaf prompt, as described in §3.

| Stage | Prompt |
|---|---|
| Construction (leaf) | `[TEXT_OF_SEGMNET]`. Summarize the above text comprehensively into a fluent passage. |
| Construction (non-leaf) | `[SUMMARIES]`. Compress each summary into a much shorter summary. |
| Navigation (triage) | The following passage(s) are the summaries of the different parts of a story.
To answer the question: `[QUERY]`
Which of the following summary is MOST LIKELY to contain information about the answer?
First provide reasoning to compare the summaries before you make the decision.
Summary 0: `[CHILD_SUMM_NODE_0]`
Summary 1: `[CHILD_SUMM_NODE_1]`
…
Summary N: `[CHILD_SUMM_NODE_N]`

Relpy with the passage number as your action.
You MUST choose one summary number and you should reply with the following format:
####################################
Reasoning: ...
Action: 0 / 1 / 2, ...
#################################### |
| Navigation (leaf) | Read the text in triple quotes and answer a question:
Story background information: `[WORKING_MEMORY]`
Main text: `[TEXT_OF_SEGMNET]`

`[QUERY]`
`[OPTIONS]`

If the answer CAN be inferred from the text above, reply with action -2, and also provide your reasoning, and the final answer.
You are ONLY allowed to reply with action -2 or -1.
Your should reply with the following format:
####################################
Reasoning: ...
Action: -2 or -1
Answer: ...
#################################### |

Table 6: Example trajectory from the SummScreenFD dataset.

| | **Navigation Trajectory** |
|---|---|
| Node 1 | The following passage(s) are the summaries of the different parts of a story.
To answer the question: *Who invited Michael to his business school as a guest speaker?*
Which of the following summary is MOST LIKELY to contain information about the answer?
First provide reasoning to compare the summaries before you make the decision.
Summary 0: In the text, Michael and Ryan are on their way to give a speech at a business school. [...]
Summary 1: Michael is reminiscing about his college days and suggests playing Frisbee with a college student. [...]
Summary 2: Michael Scott is giving a presentation to a group of business students, attempting to explain [...]
Summary 3: In the text, there is a scene where a bat is found in the office and employees react differently to its presence. [...]
Summary 4: Pam, an artist, has an art show featuring her paintings. Roy compliments her art [...]
Relpy with the passage number as your action.
You MUST choose one summary number and you should reply with the following format:
##################################
Reasoning: ...
Action: 0 / 1 / 2, ...
################################## |
| Response | Reasoning: Summary 2 provides more information about Michael giving a presentation to business students, making it more relevant to the question about who invited Michael to his business school as a guest speaker.
Action: 2 |
| Node 2 | Read the text in triple quotes and answer a question:
Story background information: Michael and Ryan are on their way to give a speech at a business school. Kevin joins them after experiencing a tire blowout on his way there. Michael is excited about [...]
Main text:
[SCENE_BREAK]
Michael: So, you wanna start a business. How do you start? What do you need? Well, first of all, you need a building. And secondly, you need supply. You need something to sell. Now this could be anything. It could be... a... thingamajig. Or a... a whosi-whatsi. Or... a Whatchamacallit . Now, you need to sell those in order to have a PayDay . And, if you sell enough of them, you will make a 100 Grand . Satisfied?
[SCENE_BREAK]
Toby: Oh, this looks great. I'd, I'd love to be there, but my daughter's play is tonight. ... Damnit! You know, one of the other parents will probably videotape it.
[...]
Question: "Who invited Michael to his business school as a guest speaker?" If the answer CANNOT be inferred from the text above, reply with action -1.
If the answer CAN be inferred from the text above, reply with action -2, and also provide your reasoning, and the final answer.
You are ONLY allowed to reply with action -2 or -1.
Your should reply with the following format:
##################################
Reasoning: ...
Action: -2 or -1
Answer: (A) ...
################################## |
| Response | Reasoning: The text does not explicitly mention who invited Michael to the business school as a guest speaker.
Action: -1 |
| Node 1
Response | [...] // Retracted to the first node.
Reasoning: Summary 0 contains information about Michael, Ryan, and Kevin going to give a speech at a business school. This summary is the most relevant to the question about who invited Michael to the business school as a guest speaker.
Action: 0 |
| Node 3 | [...]
Main text: [...]
Michael: Pop quiz.
Kevin: ...What?
Michael: Why is today a special day?
Kevin: I almost died.
Michael: Today's a special day, because I am being honored as a... visiting... professor, special lecturer, emeritus... how did you, how did you...
Ryan: You will be a guest speaker... in my Emerging Enterprises class.
Michael: In business school, Kevin. Business school.
[...] |
| Response | Reasoning: Ryan mentioned that Michael would be a guest speaker in his Emerging Enterprises class.
Action: -2
Answer: Ryan |

Table 7: Flat leaf baselines compared with MemWalker (Full Tree) on QuALITY.

|  | Flat Leafs | Full Tree |
|---|---|---|
| LLaMA-2 Chat 13B | 51.3 | 39.6 |
| LLaMA-2 Chat 70B | 51.0 | 52.0 |
| Stable Beluga 70B | 40.6 | 67.4 |

Table 8: Retrieval baseline on QuALITY with different number of retrieved chunks ($k$).

| k = 1 | k = 2 | k = 3 |
|---|---|---|
| 32.5 | 41.1 | 54.0 |

Table 9: Finish ratio.

| QuALITY | SummScreenFD | GovReport |
|---|---|---|
| 91.4 | 95.1 | 97.0 |