# OpenReview forum: "Walking Down the Memory Maze: Beyond Context Limit through Interactive Reading"
_ICLR.cc/2024/Conference — Submitted to ICLR 2024_

### Official Review · Reviewer_rkxv · 2023-10-31

**Soundness:** 2 fair
**Presentation:** 2 fair
**Contribution:** 2 fair
**Rating:** 6
**Confidence:** 4

**Summary:**

This paper proposes MEMWALKER, a method that enables the LLM to read long-text interactively via iterative LLM prompting of first creating a memory tree and second navigating this tree to answer a query. During the first stage, the long-text is segmented into small chunks that fit within the LLM's context window, to summarize into a textual summary node. These summaries are further summarized, building a hierarchical summary tree structure. MEMWALKER is evaluated on three long context question answering tasks and demonstrate superior performance to existing baselines. MEMWALKER also enhances explainability by highlighting the reasoning steps as it interactively reads the text and can pinpoint the relevant text segments related to the query.

**Strengths:**

* The paper is clearly written and easy to follow.
* The paper tackles a real world and important problem of question answering on long input documents that are beyond the context window constraint of LLM-based LLMs.

**Weaknesses:**

* One limitation of the method is that each time the underlying long input changes, the memory tree must be constructed again, which requires processing the entire long input multiple times through an expensive LLM. This is more than just processing all the input tokens, as the hierarchical summaries are also created to form the *memory tree*. Therefore it could be useful to include a discussion of this limitation, and perhaps due to this, this method is best suited to applications where the long input does not change, such as customer service question answering for pre-defined product information.
* It could be useful for the reader if you could provide more empirical results with other LLMs, e.g., GPT3.5, GPT4, Llama 2 etc.
* It could be helpful for the reader if you provide practical guidelines for setting the maximums number of nodes, and the segment size for a given new task or practical example.
* No error bars for results. Could you include error bars for the results in Table 2, Table 3, Figure 2, Figure 3, Table 4, Figure 4 and Figure 5. Given there are no error bars, the results seem marginal, particularly for Table 2, and it appears the method is only performing significantly well on the GovReport task. Including error bars would help the reader see which results are statistically significant and not overlapping.
* For Contriever (the retrieval baseline), it would be more indicative if you only included the top k retrieved responses, where $k=3$ for example or even less, instead of including all segments until they fill the context (as this could bloat the context with irrelevant information---potentially leading to worse performance).
* Figure 4 is misleading, as the total tokens to process, would be greater than the original example in tokens, as the original example needs to be initially processed into the *summary tree*.







Typos:
* Page 1. "consuming large amount of" -> "consuming large amounts of"
* Page 2. "find the node that contains relevant" -> "find the node that contains a relevant"
* Page 7. "into two bucket of" -> "into two buckets of"

**Questions:**

* For future work, it could be helpful to discuss the extension of using the query, or types of expected queries to construct the *memory tree*. As I can imagine knowing the types of queries or the family of possible queries that will be asked, the *memory tree* could be constructed for that particular query family, to give specific summarizations that best address those families of queries, and enable efficient searching.
* *" If the LLM fails to generate parsable output three consecutive times, the navigation terminates and returns “no answer”."*. How many times does this occur in practice?
* *"Further summarizing the working memory as it accumulates would be an alternative approach, which we
have not explored in this study."*. It could be really interesting to explore this as a further ablation.
* Can the method handle a larger maximum number of nodes $M_t$? Can you perform an ablation, perhaps showing it handling $M_t =\\{ 16, 32, 64, 128\\}$?
* In Table 4, the stray ratio is quite high, is there any way to lower it? Does this indicate that the method is not the most efficient approach?

---

> ### Author Response · Authors · 2023-11-21
> **Thank you for the feedback**
>
> Thank you for the feedback! Please let us know if you have further questions.
>
> **Volatility of the memory tree**
>
> Thank you for bringing up this interesting point! Our perspective to this question is twofold: 1) our targeted use cases are for static texts where the memory generation stage only needs to be performed once per document, which might be suitable in many question answering settings, and 2) the entire memory tree does not need to be regenerated. If a node is modified, only the path leading towards that leaf node needs to be regenerated. This warrants only O(logN) updates to the original memory tree (N=total number of nodes).
>
> **Assessment on other LLMs**
>
> We provide comparison to LLaMA 2 chat of size 13B and 70B in Table 2 and 3. We are happy to provide comparisons on GPT-4 and Claude, but we do caution that due to opacity of these systems, the results might suffer from issues like data contamination. We are happy to provide extra experiments on these models for reference in the updated version.
>
> **Setting k=3 or other values for the retrieval baseline**
>
> We provide the new retrieval baselines results as suggested on QuALITY:
> | k = 1 | k = 2 | k = 3 |
> |-----------------|-----------------|-----------------|
> |  32.6 |  41.1 |  54.0 |
>
>
>
> Results show a consistent trend that increasing the number of retrieved chunks improves performance. The issue of distraction by irrelevant content has not been present in our setting. We added the suggested experiments in the Appendix.
>
>
> **Finish ratio**
>
> | QuALITY | SummScreenFD | GovReport |
> |-----------------|-----------------|-----------------|
> |    91.4 |         95.1 |      97.0 |
>
> We provide finish ratios on each dataset where the model is unable to parse its response 3 consecutive times. On all datasets, finish ratios are above 90% and are not the major source of error in MemWalker. We added the experiments to the Appendix.
>
> **Maximum number of nodes**
>
> The maximum number of nodes is restricted by the maximum allowable context length. For example, if each node has text of length X tokens, and the maximum context window for input from the LLM is C, the maximal number of nodes that can be connected to a parent is min’(M_t, C / X). This means that even if we set a high M_t value, it will be bottlenecked by the LLM context length. Setting M_t to 16 reaches the limit given the average summary length X per node, therefore increasing M_t wouldn’t change the tree structure.
>
> **Improving stray ratio**
>
> We expect the stray ratio to improve as the reasoning capability improves. This can be achieved by better zeroshot prompting techniques, fine-tuning on navigation paths, or simply deploying a stronger LLM.
>
> **Significant tests**
>
> We will provide significance test statistics in the updated version.

---

> > ### Comment · Reviewer_rkxv · 2023-11-22
> > **Reviewer Response**
> >
> > Thank you for the answers provided and the additional experiments. They have mostly addressed my concerns. However, I would still like to see full-significance test statistics in the updated version. I have increased my score, conditional on significance test statistics being included, that is, error bars for all results, including the figures.

---

> > > ### Author Response · Authors · 2023-11-22
> > > **Thank you**
> > >
> > > Thank you for the updated assessment! We appreciate the feedback and will include the promised tests to the updated version.

---

### Official Review · Reviewer_T9FN · 2023-11-01

**Soundness:** 1 poor
**Presentation:** 3 good
**Contribution:** 2 fair
**Rating:** 3
**Confidence:** 4

**Summary:**

The authors propose a method for handling long texts using LLMs. The method is based on first building a hierarchical summary tree and then, given a query, traversing the tree in order to find the relevant part of the input document. The authors perform a number of experiments that demonstrate the promise of their method, especially on long sequences.

**Strengths:**

The paper
- Attacks a highly relevant topic of handling extremely long documents with LLMs which might not have long enough context to handle such sequences.
- Proposes a new approach to the problem.
- Is clearly written and is a pleasure to read.

**Weaknesses:**

Overall, I could see that the authors put a lot of work in their paper, and I appreciate the contribution in many ways, I believe that unfortunately, the paper will require substantial improvement before it (in my opinion) could satisfy the extremely high bar of the ICLR conference. The main weakness of the paper is the quality of its experimental support. Novelty is another concern.

## Experimental support issues.

### Retrieval comparisons
The paper, fundamentally, proposes a way to work around context length limitations through a creative use of summarization. It is crucial, therefore, to provide thorough comparisons with other methods of this type. We only have a comparison with Contriver, which is a BERT-based architecture. Given that the underlying model in Contriver is much weaker than what was used in MemWalker, it is impossible to evaluate what proportion of performance gains was due to using MemWalker summarization technique and what proportion is due to a stronger base model.

To provide a fair comparison, it seems crucial to find a base model that could be used for both MemWalker and retrieval-augmented approaches. Any LLM that can output sentence/paragraph embeddings and is instruction fine-tuned could be a good start.

### Simple summary comparisons

Another comparison that seems to be missing is a non-tree-based summary search. I understand that tree-based search speeds up query answering, but sometimes we might only have one query per document. It seems important to compare MemWalker with a simple flat summary scanning approach (Figure 5 looks promising, but still only considers tree-structured approaches).

### Long-Context LLM comparisons

On inputs that fit into LLM context windows, we don't see any advantage of MemWalker over the base model. Truncating the context would, of course, degrade performance.

The paper states that scaling context lengths is fundamentally limited ("the attention mechanism may become less effective due to positional biases as the sequence length becomes very long"). I agree that it might be the case, but to show that MemWalker addresses the issue, we need to have a fair comparison, that is, we need to compare MemWalker with models that can fit the inputs into their context.

One simple option would be to add GPT-4 (long-context version) and/or Anthopic's Claude. Both models would be able to fit even the longest sequences considered in the paper into their context.

To make the case clearer (to push context scaling to its limit), it could be reasonable to focus on longer-context documents than what was done in the paper. [NarrativeQA](https://direct.mit.edu/tacl/article/doi/10.1162/tacl_a_00023/43442/The-NarrativeQA-Reading-Comprehension-Challenge), BookSum (https://arxiv.org/abs/2105.08209), and NarrativeXL (https://arxiv.org/abs/2305.13877) could all be a good place to start.

### Experiment scale and result reporting

In general, the authors used very small data subsets to evaluate model performance. I understand the resource limitations involved in such studies. But, especially when data is so limited, it is crucial to provide a thorough statistical analysis of the obtained results. As presented, it is impossible to estimate which results reflect actual differences in model performance and which are due to noise.

## Novelty concerns

While I understand that the specific way in which the model is traversing the memory tree is novel and original, using hierarchical summarization as a way to handle long documents is an idea that has been known for a long time.

See, e.g. Yang, Christopher C., and Fu Lee Wang. "Hierarchical summarization of large documents." Journal of the American Society for Information Science and Technology 59.6 (2008): 887-902.

While this limits the novelty and potential significance of the work, this limitation is not as crucial as the experimental support issues I listed above.


## Conclusion

Overall, unfortunately, in its present state, I can not recommend the paper for publication. I do hope to see a revised version of this paper published in the future.

**Questions:**

In case I missed it, I was wondering what was the performance of a "flat summarization" approach (i.e. no tree, only leaf-level summaries + search over all of them during retrieval). I understand that some relevant data is reported in Figure 3, but it seems that "flat summarization" and "mem-walker w/o memory" are not exactly equivalent.

---

> ### Author Response · Authors · 2023-11-21
> **Thank you for the feedback**
>
> Thank you for the feedback! Please let us know if you have further questions.
>
> **Suitability of Contriever as the retrieval baseline**
>
> We chose Contriever because it was one of the SoTA document level retrieval models available at the time we conducted the experiments. Despite basing on BERT as the base model, it is trained to perform well specifically on retrieval tasks, which might not be necessarily weaker than the LLM performing reasoning on a specific node.
>
> **Flat summary baselines**
>
> Thank you for the suggestion on the flat summary ablation. We have now run the suggested experiments and provide results on the QuALITY dataset:
>
> | Model   |   Flat Leafs | Full Tree |
> |-----------------|-----------------|-----------------|
> | LLaMA-2 Chat 13B  |       51.3 |      39.6 |
> | LLaMA-2 Chat 70B  |       51.0 |      52.0 |
> | Stable Beluga 70B |       40.6 |      67.4 |
>
>
> We prompt the models to first generate whether the answer can be inferred from the summary leaf node. If the answer is yes, then keep generating the answer. We take the majority vote across all generations to determine the final answer. We observe that the flat summary approach gives a pretty unstable result. This is mainly due to ill-calibration, i.e., the model tends to predict overwhelmingly yes or no given the text (Stable-Beluga 70B gives overwhelmingly no as answer). This suggests that using the tree structure is necessary in the zero-shot scenario. We added the suggested experiment to Appendix.
>
> **Long context models & Longer length datasets**
>
> Please refer to the common response: *Choice of long context LLMs & baselines*
>
>
> **Novelty comparing to hierarchical summarization**
>
> Thank you for bringing up this point. We recognize that the leveraging hierarchical structure for documents has a long history by including the discussion of work by Wu et al., 2019 [1]  that leverages hierarchical structure for summarization in the related work discussion. We will include the mentioned reference into the updated version! Overall, the method aimed to summarize the document from the ground up, sharing resemblance to the first stage in MemWalker. A major difference is that in MemWalker, each node only needs to contain sufficient information to triage the LLM into the children nodes, while in Wu et al., 2019, the challenge was to ensure that the root node is a faithful summary of the entire document. In addition, to our knowledge, we are the first to allow the LLM to navigate in the tree structure to answer questions.
>
> [1] Jeff Wu, Long Ouyang, Daniel M. Ziegler, Nisan Stiennon, Ryan Lowe, Jan Leike, and Paul Christiano. Recursively summarizing books with human feedback.
>
> **Significant tests**
>
> We will provide significance test statistics in the updated version.

---

> > ### Comment · Reviewer_T9FN · 2023-11-23
> > **Thank you for your response**
> >
> > I've read the authors' response as well as other reviews. I appreciate that the authors thoughtfully engaged with the reviews.
> >
> > Unfortunately, however, I still stand by my original assessment.
> >
> > When it comes to the "flat tree" baseline, the observed issue (the model overwhelmingly responding "yes" or "no") could be potentially mitigated with further prompt engineering (e.g. asking to output confidence rather than yes/no). If there is access to logits, one can directly take the confidence associated with "yes" and "no" potential continuations. Then the highest-relevance segment can be selected. Even setting this issue aside, we see that for some models (LLaMA-2 Chat 70B), the proposed method and the flat search perform nearly equivalently. I can't say that these results provide strong support for the method (i.e. it helps some models but not others).
> >
> > Overall, the method has substantial limitations when it comes to scalability and novelty. Moreover, the paper's experimental support, in my view, is not strong enough to demonstrate that the method will be widely useful to the community. The authors re-frame the contribution as a universal high-level add-on to any LLM, but even in the small additional experiments, we see that some LLMs do not substantially benefit from it.
> >
> > I am sorry that I can not at present increase the score. I hope that in the future, the authors could run more comprehensive evaluations and comparisons, and, if the method reliably offers improvements, I hope to see this work published in the future.

---

### Official Review · Reviewer_yntC · 2023-11-01

**Soundness:** 2 fair
**Presentation:** 2 fair
**Contribution:** 2 fair
**Rating:** 5
**Confidence:** 4

**Summary:**

This paper proposes MemWalker, a method that allows LLMs to process long context corpora through iterative prompting. It does by first 1) constructing memory tree- text is recursively summarized into a tree structure with summary nodes, going from segments to higher level summaries. 2) Navigating the tree- Given a query, the LLM navigates the tree structure by reasoning about which child node to go to in order to find relevant information, reaching a leaf node containing the full text segment that allows it to answer the query.

**Strengths:**

The paper outperforms baselines on 3 datasets and shows good results. The navigation process of the memory tree gives some explanation of the model's reasoning process of the answer. The idea is interesting.

**Weaknesses:**

The scalability of MemWalker is a concern. The method doesn’t outperform full context for very short texts. It is unlikely that the model will scale to extremely large scale texts given the computational overhead, limiting the model to medium length texts. As the context windows of models expand, many medium-sized documents will fit in the context length, making the MemWalker not necessary.

The comparison with two large-context models using a significantly smaller 13B parameter set against MemWalker's 70B model does not seem fair. A  better comparison would be with models of similar size, for example with  llama models fine-tuned for larger contexts using positional interpolation.

Details on the time and token scaling of MemWalker are absent thus it's unclear if the computational costs are worth the increased quality of medium-sized document. MemWalker bears a resemblance to the tree index method by the llama index and the description of the method in the paper is not sufficient (please see questions).

The method also relies heavily on summarizations which may suffer from hallucinations, and there is no discussion of this.

**Questions:**

How are the nodes grouped together to be summarized ?

In the case the grouped nodes exceed the maximum context length of the model ? how is this case handled ?

How does the method scale in terms of wall clock and tokens ?

How does the model compare to larger context windows with model of similar size ?

---

> ### Author Response · Authors · 2023-11-21
> **Thank you for the feedback**
>
> Thank you for the feedback! Please let us know if you have further questions.
>
> **Limitation on extremely long sequences**
>
> Please refer to the common response.
>
> **Choice of long context LLMs & baselines**
>
> Please refer to the common response.
>
> **Discussion on llama index**
>
> Thanks for pointing this out! We recognize that MemWalker shared similar motivation and certain approaches to llama index (Tree Index in particular). As far as we know, llama index targets practitioners who aim to use LLMs as building blocks to quickly build applications, which does not aim to perform comprehensive evaluation and analysis on shared academic benchmarks. Our analysis on the reasoning capability threshold and the necessity of working memory go beyond simple tree building and querying and focuses on how this leads to better models to understand long context. We will add the discussion to the related work section in the updated version.
>
> **Hallucination in the generated summaries**
>
> Thank you for pointing this out – we agree that generative approaches run the risk of hallucination. However, in our use case, the purpose of summary nodes are only meant to triage the LLM into different leaf nodes. The final answer will depend on the actual content of the document. The level of hallucination will indeed impact the navigation stage and make LLM take detours more often. Although we cannot control the amount of hallucination in the generated summary, it would be a beneficial follow-up analysis to this line of methods and add irrelevant content in the summary nodes and measure the impact of the navigation stage.
>
> **Questions on node grouping**
>
> The context window of the LLM is first separated into two parts: input and generation. When the contents of the nodes are grouped into a parent node, we group them by 1) the predetermined max number of nodes, and 2) only grouping nodes where their concatenated length is smaller than the input part of the LLM context window. This ensures that the LLM can always summarize the children nodes into the parent node.

---

### Official Review · Reviewer_nCny · 2023-11-02

**Soundness:** 3 good
**Presentation:** 2 fair
**Contribution:** 3 good
**Rating:** 8
**Confidence:** 4

**Summary:**

In the research article titled "Walking Down the Memory Maze: Beyond Context Limit through Interactive Reading," the authors address the challenges associated with long-sequence question-answering tasks using Large Language Models (LLMs). Given that these tasks often require referencing extensive text segments that surpass the standard context-window of an LLM, the study introduces an alternative to extending the LLM's context-window or incorporating recurrence or retrieval-augmented generation. The proposed method involves segmenting and summarizing the text, followed by the assembly of a navigable knowledge tree composed of these summaries. Importantly, the leaf nodes of this tree retain the original text segments, enabling the model to interactively traverse and reference the content, enhancing its question-answering capabilities.

**Strengths:**

***Strengths of the Paper:***
1. Novelty and Originality:
The paper introduces MemWalker, an alternative approach to managing long-sequence reading tasks, suggesting a new direction that differs from the existing trend of expanding model context windows.

2. Rigorous Experimental Design:
The paper's experimental framework is detailed, employing a variety of datasets and metrics, which supports the validity of the MemWalker system's evaluated performance.

3. Clarity of Presentation:
The methodological process, including memory tree construction and interactive navigation, is clearly delineated, facilitating understanding and potential replication of the study.

4. Substantial Findings:
MemWalker's performance across different datasets and tasks is solid, indicating its usefulness in question-answering and information retrieval within long texts.

5. Detailed Analysis and Discussion:
A comprehensive analysis is coupled with a discussion that explores the broader implications and areas for future exploration, placing the findings in context with the existing body of research.

6. Quality of Writing:
The paper is well-composed, with content organized in a manner that aids in the clear communication of the research to the reader.


-----

***Strengths of the Approach:***

1. No Need for Fine-tuning:
Unlike other methods, MemWalker avoids the cost-intensive process of extensive fine-tuning for longer sequences.

2. Retention of Older Sequence Information:
MemWalker is designed to retain older sequence information, avoiding the typical compression seen in some recurrent architectures that weakens recall.

3. Effective Navigation with Working Memory:
The inclusion of working memory in MemWalker bolsters its performance, demonstrating its essential role in the navigation process.

4. Ability to Recover from Traversal Errors:
MemWalker exhibits some level of resilience, demonstrative effective recovery from initial navigation errors that lead to improved accuracy.

5. Efficient Content Reading:
MemWalker's methodology enables it to read and process content efficiently, requiring a smaller portion of the entire text to derive accurate answers.

6. Flexible Memory Tree Construction:
MemWalker's approach to memory tree construction strikes a balance, providing for some level of information compression without sacrificing content fidelity.

**Weaknesses:**

***Shortcomings of the Paper:***

1. The results would be more convincing if they were compared against the SOTA retrieval, recurrence and length-extention tuned models. Currently, all results (except length-exntention fine-tuning) are based on Beluga 2 which is a the main weakness of this paper.

2. It would be good to see how this method compares to recent approaches such as Landmark Attention: https://arxiv.org/abs/2305.16300

3. The lack of a broader impact section weakens this paper.

----
***Shortcomings of the Approach:***
1. Scalability with Extremely Long Sequences:
MemWalker's memory tree generation might struggle with scalability for extremely long sequences. The growth in sequence length could lead to an overwhelming number of nodes, complicating the tree construction process.

2. Dependency on LLM's Reasoning Capability:
MemWalker's effectiveness is deeply tied to the robust reasoning capabilities of the LLM. The system requires a large (over 70B) and instruction-tuned LLM. A deficiency in this capability could compound errors, jeopardizing the method's success.

3. Limitation of Zero-Shot Prompting:
MemWalker solely relies on zero-shot prompting without tapping into the potential benefits of fine-tuning. This could constrain its interactive reading capabilities, leaving room for enhancement.

-----

These are all shortcomings of the approach, highlighted and recognised by the authors.

**Questions:**

Will code be open-sourced (including the evaluation code and data splits)?

Is it possible to provide results for benchmark on even longer contexts, e.g. 30k+ tokens?

-----

**Improving clarity and grammatical ammendments:**

***Section 1. Introduction***

Original: "These tasks involve consuming large amount of information,"

Proposed Correction: "These tasks involve consuming a large amount of information,"

-------------

Original: "The context window, no matter how long it is extended, assumes a fixed size,"

Proposed Correction: "Regardless of how it is extended, the context window assumes a fixed size,"

-------------

Original: "While recurrence can manage infinite-length sequences, it often misses out on retaining information from earlier segments."

Proposed Correction: "Although recurrence can handle infinite-length sequences, it frequently fails to retain information from earlier segments."

-------------

Original: "Additionally, retrieving segments from the coherent long-text might be ineffective, given that many retrieval systems are tailored to distinguish similar but distinct documents."

Proposed Correction: "Furthermore, retrieving segments from coherent, extended texts might be ineffective since many retrieval systems are designed to differentiate between similar yet distinct documents."

-------------

Original: "To address these issues, we develop a fundamentally different approach which treats the model with a finite context window as an interactive agent,"

Proposed Correction: "To address these issues, we introduce an approach that treats the model with a finite context window as an interactive agent,"



***Section 2. Related Work***


Original: "Another direction is modified self-attention."

Proposed Correction: "Another approach involves modifying self-attention."

-------------

Original: "to enable models on longer sequences"

Proposed Correction: "to enable modelling on longer sequences"

-------------


Original: "Despite the recent advances, this approach comes with two natural limitations:"

Proposed Correction: "Despite recent advancements, this method presents two inherent limitations:"

-------------

Update: Authors addressed some of my concerns.

---

> ### Author Response · Authors · 2023-11-21
> **Thank you for the feedback**
>
> Thank you for the feedback! Please let us know if you have further questions.
>
> **Limitation on extremely long sequences**
>
> Please refer to the common response.
>
> **Suitability of Stable-Beluga based baselines**
>
> Please refer to the common response *Choice of long context LLMs & baselines*.
>
> **Comparison to SoTA recurrence & retrieval methods**
>
> We aim to construct baselines by sharing as many comparable components as possible to our main method. Therefore, the recurrence baseline uses Stable Beluga 70B in a similar zero-shot prompting fashion despite other recurrent methods available that train on extra datasets. We chose Contriever as it was one of the best retrieval models at the time of development of MemWalker. Similarly, the final prediction is performed by the same Stable Beluga 70B model to keep components comparable.
>
> **Typos & suggested writing changes**
>
> Thanks for pointing them out. We will fix them in the updated version.

---

### Author Response · Authors · 2023-11-21
**General Response**

We thank all reviewers for their time and feedback! We have revised the draft to incorporate your comments (in blue).

Below are the general response to common concerns and questions raised by more than one reviewer.

**Re-emphasis of contribution**

We would like to highlight that MemWalker is not tied to any specific LLM. In fact, the stronger the LLM’s reasoning capability or the longer the permissible context window, the more effective the approach is. We demonstrate that MemWalker provides benefits after an LLM passes a certain reasoning capability and length threshold. We expect stronger models to further this benefit.

**Limitation on extremely long sequences**

Several reviewers mentioned MemWalker’s limitation to extremely long sequences. We agree that scaling is a potential issue when handling extremely long text during memory tree generation. However, this scaling issue could be mitigated as the underlying LLM permits longer context length. For example, given a document of length L, an LLM with a maximum of C tokens, the memory tree consists of N = L / C leaf nodes. As C increases with more capable LLM, N decreases, resulting in a smaller memory tree.

**Choice of long context LLMs & baselines**

Several reviewers have raised questions about further experiments on LLMs such as GPT-4, Claude, where the model fits the entire input sequence. We agree that in order to fairly compare, evaluating a model that can fit the entire context would be desirable. However, we are limited by the largest open long context models currently available: 1) closed models such as GPT-4 or Claude would provide a reference point, but due to their opacity it might run into issues such as data contamination, and 2) the only suitable alternative is LLaMA 2 Long 70B (paper arxived after MemWalker), which is currently unavailable for us to test. The other available long context alternatives are smaller in size.

---

### Meta-Review · Area_Chair_cCiu · 2023-12-10

**Metareview:**

This paper presents MemWalker, a method that enables LLMs to process extensive contexts efficiently. It operates through two main steps: (a) constructing a memory tree by recursively summarizing documents into a hierarchical tree structure featuring summary nodes, and (b) enabling LLMs to navigate this tree in response to a query. This navigation leads to a leaf node containing the detailed information required to answer the query.

The paper is well-structured and shows reasonable improvements across three benchmarks. However, most reviewers have expressed concerns about the scalability of MemWalker. These concerns include: (a) the method's underperformance compared to full context analysis for short texts; (b) potential scalability issues in generating memory trees for extremely long sequences; (c) the redundancy of MemWalker as LLMs evolve to accommodate medium-sized documents within their expanding context windows.

While the authors argue that scaling issues might be mitigated with the advancement of LLMs capable of handling longer contexts, this argument inadvertently highlights the proposed method's diminishing relevance. It appears to be a temporary solution pending the development of LLMs with even longer context windows. After a thorough review of the rebuttal and subsequent discussions, I am inclined to recommend a rejection of the paper due to these concerns about its long-term impact

**Justification For Why Not Higher Score:**

See above regarding the scalability concern and potential impact of the work.

**Justification For Why Not Lower Score:**

N/A

---

### Decision · Program_Chairs · 2024-01-16

Reject